# Spatial Heterogeneity Analysis of Short-Duration Extreme Rainfall Events in Megacities in China

**Qi Zhuang** [1], **Shuguang Liu** [1,2] **and Zhengzheng Zhou** [1,*]

[1] Department of Hydraulic Engineering, Tongji University, Shanghai 200092, China; 1932631@tongji.edu.cn (Q.Z.); liusgliu@tongji.edu.cn (S.L.)

[2] Key Laboratory of Yangtze River Water Environment, Ministry of Education, Tongji University, Shanghai 200092, China

[*] Correspondence: 19058@tongji.edu.cn

**Abstract:** Given the fact that researchers require more specific spatial rainfall information for storm flood calculation, hydrological risk assessment, and water budget estimates, there is a growing need to analyze the spatial heterogeneity of rainfall accurately. This paper provides insight into rainfall spatial heterogeneity in urban areas based on statistical analysis methods. An ensemble of short-duration (3-h) extreme rainfall events for four megacities in China are extracted from a high-resolution gridded rainfall dataset (resolution of 30 min in time, $0.1° \times 0.1°$ in space). Under the heterogeneity framework using Moran's I, LISA (Local Indicators of Spatial Association), and semi-variance, the multi-scale spatial variability of extreme rainfall is identified and assessed in Shanghai (SH), Beijing (BJ), Guangzhou (GZ), and Shenzhen (SZ). The results show that there is a pronounced spatial heterogeneity of short-duration extreme rainfall in the four cities. Heterogeneous characteristics of rainfall within location, range, and directions are closely linked to the different urban growth in four cities. The results also suggest that the spatial distribution of rainfall cannot be neglected in the design storm in urban areas. This paper constitutes a useful contribution to quantifying the degree of spatial heterogeneity and supports an improved understanding of rainfall/flood frequency analysis in megacities.

**Keywords:** extreme rainfall; short duration; spatial heterogeneity; megacities

## 1. Introduction

Extreme rainfall is a crucial driver of flood and waterlogging, especially in urban settings where there is a fast hydrological response and high catchment variability [1–3]. With climate warming [4] and rapid urbanization [5], there is an increasing trend in the occurrence or intensity of extreme rainfall events in many areas, including China [6–9]. Heterogeneity in the land use and infrastructure [10,11] will also contribute to show distinctive urban-rural differences of rainfall in space, however, in the commonly used design storm/flood practices, the spatial representation of rainfall is often simplified or ignored [12,13], increasing uncertainty in hydrological analysis and prediction [14–16].

More strong evidence shows that spatial heterogeneity has quite an impact on the hydrologic response [17]. Both peak flow and runoff volume can be influenced by different rainfall distribution patterns [18,19]. Segond [15] found that spatial heterogeneity of rainfall could decrease the performance of the urban runoff model. This finding was also supported by Zoccatelli et al. [20]. Regarding flood response, Zhu et al. [21] even found that rainfall spatial structure is more important than a temporal structure for drainage areas larger than approximately 2000 km². Thus, more specific analysis in rainfall spatial heterogeneity is required for hydrological analysis in cities [22–24].

Affected by topography [25], land-water boundaries [26], or land-use types [17], the differences in the rainfall spatial variability have been observed both at the global scale [27] and the regional scales [17,28,29]. Patel [30] examined the seasonal and annual rainfall extent in space based on spatial seasonality index and precipitation concentration index. the Rainfall-Weighted Flow Distance (RWD) was introduced by Smith [31] to describe the locations of the rainfall centroids in an urban watershed. In Guo's study [32], the spatial distribution of rainfall in Beijing was numerically investigated by using Mesoscale Modeling. Zhou [33] present an investigation of rainfall heterogeneity and its consequences for rainfall frequency analysis using storm catalogs combined with stochastic storm transposition (SST). Zhang [34] proposed a coefficient of variance and Moran's I to classify rainfall variabilities in the Brue catchment, UK. Despite these efforts, the spatial heterogeneity of extreme rainfall in urban areas remains poorly understood.

The main challenge faced by many researchers is how to represent and quantify the degree of rainfall spatial heterogeneity. At the cost of lacking a quantitative evaluation of rainfall spatial heterogeneity, it is difficult to establish a quantitative relationship between impact factors and spatial heterogeneity. It is also difficult to further apply the results of rainfall spatial heterogeneity in improving the accuracy of hydrological calculations. Resolving this issue would constitute a major step toward matching hydrological model performance/design storm with rainfall spatial variability [33,34]. At present, most studies have only mapped the rainfall to show the spatial heterogeneity in general [35], or analyzed the trend features in different locations over the study areas [36–38], using some conventional indices, such as annual mean precipitation, consecutive wet days, annual total precipitation when daily precipitation >95th percentile, and so on. These studies focus on temporal variations at the same locations, and the spatial distribution is represented by spatial interpolation based on results in more locations. Although they help to shed new light on precipitation changes in different rainfall stations across the study areas, the relationship of rainfall between different locations has seldom been considered.

The objective of this paper is to build a comprehensive framework to analyze and quantify the spatial heterogeneity in extreme rainfall in urban areas. Considering that rainfall in adjacent geographical locations may have similar characteristics, many indicators in geostatistics can be used to describe this heterogeneity. The spatial clustering structures of both rainfall magnitudes and locations are considered by combining CV and Moran's I [34,39,40]. The specific locations of rainfall clusters are identified by LISA [41] with decomposing Moran's I into each grid cell. The semi-variance analysis [42] is applied to study the range of rainfall clusters even the rainfall variability in a different direction in space. Based on that, we also try to analyze the influence of urban growth on rainfall spatial heterogeneity and the influence of spatial heterogeneity on estimating rainfall frequency.

The rest of this paper is organized as follows. Section 2 introduces the study areas and data. Section 3 presents the methods used in this study. The temporal characteristics of the annual maximum 3-h rainfall events (Rx3h) are analyzed in Section 4.1. The area mean of Rx3h is taken to analyze spatial clustering in Section 4.2.1. The locations of rainfall centroids in each Rx3h event are illustrated in Section 4.2.2. To increase the sample size and reliability of the results, the annual top 10 3-h rainfall events are taken into consideration for annual spatial clustering in Section 4.3.1. The influence of spatial heterogeneity on rainfall frequency analysis is discussed in Section 4.3.2. Section 5 presents a summary and conclusions.

## 2. Study Areas and Data

### 2.1. Study Areas

This study focuses on four megacities in China including Shanghai (SH), Beijing (BJ), Guangzhou (GZ), and Shenzhen (SZ). These cities are economically the top four cities in China and are known as famous global cities, where the population and urbanization have tremendous changes over the past decades. Though they have experienced substantial urban development, the risk

of rainfall-induced urban flooding remains a major problem for these cities, with the 21 July 2012 flooding [43] in BJ associated with extreme storms providing the most notable example.

The details of the characteristics of SH, BJ, GZ, and SZ are shown in Table 1. Generally, impacted by their geophysical locations, GZ and SZ have a larger amount of annual rainfall (1915 and 1801 mm) than BJ (441 mm). SH, as a coastal city, lies at the mouth of the Yangtze River where abundant moisture can be transported to it, with a prevalent southeasterly wind from its adjacent seas. This area is frequently affected by cyclonic storms and intense convectional precipitation especially during the flood season (June to September) [35,44]. BJ has a warm monsoon climate which is characterized by large seasonal variations in temperature and precipitation. It is characterized by mountain-plain topography and the continental monsoon climate, resulting in an extremely un-uniform spatial distribution of precipitation [11,45]. Situated on the coast of South China, GZ and SZ are affected by subtropical maritime monsoon with high temperatures and continuous rainfall. Extreme rainfall events triggered by monsoons and typhoons are often composed of multiple short-duration rainstorms with short intermissions in GZ and SZ [46]. It should be noted that the islands in SH and SZ are excluded from this study. Figure 1 shows the geographical location, terrain types, and land cover types of each study city.

**Table 1.** Information about the four study cities.

| Cities | SH | BJ | GZ | SZ |
|---|---|---|---|---|
| Areas (km²) | 6340.5 | 16,410.54 | 7434.4 | 1997.47 |
| Terrain types | Plain | Mountains, and plain | Hilly land | Hilly land, and plain |
| Elevation (Datum: WGS_1984, m) | 0–101 | −121–2306 | 0–1185 | 0–936 |
| Climate | Subtropical monsoon | Continental monsoon | Subtropical maritime monsoon | Subtropical maritime monsoon |
| Annual average temperature (°C) | 17.7 | 12.6 | 22.3 | 22.4 |
| Annual rainfall (during 2000–2020, mm) | 1320 | 441 | 1915 | 1801 |
| Population density (people in per square kilometer) | 3830 | 1312 | 2059 | 6728 |
| Urbanization rate (data up to 2019) | 88.1% | 86.6% | 86.1% | 99.7% |
| GDP (billion CNY) | 3815.53 | 3537.13 | 2362.86 | 2692.71 |

### 2.2. Data

Rainfall data from the satellite gridded dataset are employed to analyze spatial heterogeneity in urban areas. The dataset is obtained from the Global Precipitation Measurement (GPM) mission (https://gpm.nasa.gov/data/directory), with high-quality control, and has been widely used in many previous studies [47,48]. The spatial resolution of the gridded rainfall data is 0.1° × 0.1° (about 10 km × 10 km). and the temporal resolution is 30 min. The period used in this study is from 2000 to 2020.

The land use maps in 2001, 2015, and 2017 are obtained from the GlobalLC_MOD_2001 [49,50] data set and the FROM-GLC30 data set (http://data.ess.tsinghua.edu.cn/). The map has a resolution of 30 m × 30 m, reflecting the change in land cover from 2001 to 2017.

We have examined the time series of annual maximum rainfall events with time scales from 1-h, 3-h, 6-h, to 12-h. Considering the temporal resolution of the data (30 min) and its pronounced increasing trend, 3-h annual maximum rainfall events are extracted to represent the short-duration rainfall events.

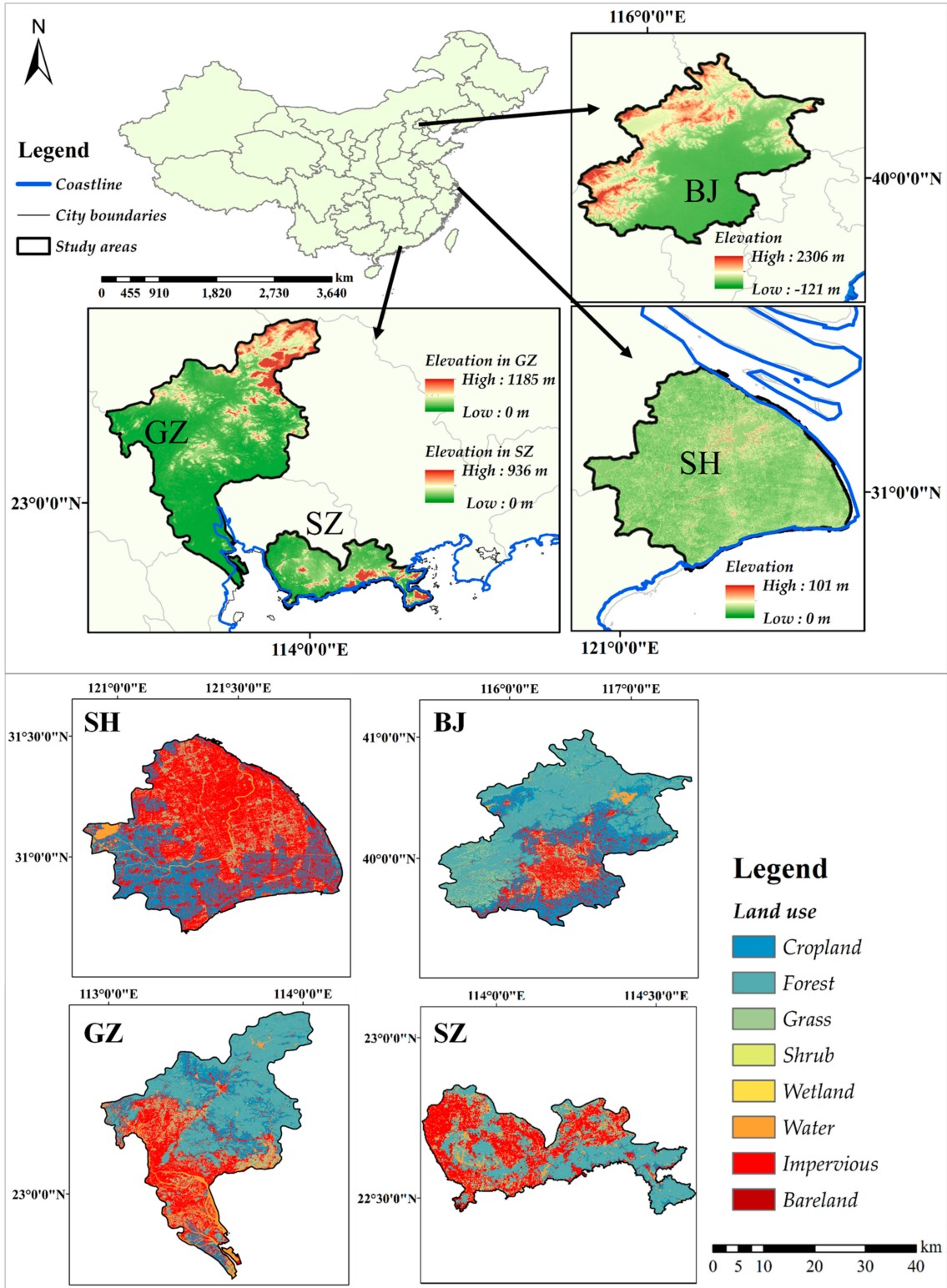

**Figure 1.** Location maps of the study areas with topography, city boundaries, and land use.

## 3. Methodology

### 3.1. Trend Analysis

The Mann–Kendall (M-K) trend test [51,52] has been used to detect the trend and significance of hydrological series, which is highly recommended by many researchers [53,54]. For the time series of observations $X = \{x_1, x_2, \cdots x_T\}$, the statistic $S$ is calculated in Equation (1):

$$S = \sum_{i=1}^{T-1} \sum_{j=i+1}^{T} sgn(x_j - x_i) \tag{1}$$

where $T$ is the length of the series, and the "*sign*" function is computed as:

$$sign(x_j - x_i) = \begin{cases} 1 & if\ (x_j - x_i) > 0 \\ 0 & if\ (x_j - x_i) = 0 \\ -1 & if\ (x_j - x_i) < 0 \end{cases} \tag{2}$$

If $S$ is greater than 0, the series is considered to have an increasing trend, otherwise, it is considered to have a decreasing trend.

To further estimate the slope of trend, Sen's slope method [55], a non-parametric procedure develops for the time series of observation $X = \{x_1, x_2, \cdots x_T\}$. The linear variation of the mean is expressed by $\beta$:

$$\beta = Median\left(\frac{x_j - x_i}{j - i}\right),\ \forall i < j \tag{3}$$

in which "*Median*" is the function that takes the median of the series.

Sen's slope estimator has been widely used in hydro-meteorological time series [54,56], which can further obtain the estimated value of linear trend degree based on the Mann–Kendal method.

### 3.2. Spatial Heterogeneity Analysis

#### 3.2.1. CV in Space

The coefficient of variance (CV) is the most common statistical method to show the degree of deviation between the variable and the mean. It is defined in Equation (4). A large CV refers to the presence of the striking variation within the rainfall. The advantage of CV is direct and simple to describe the spatial variability of all data, while the spatial distribution of rainfall data is neglected.

$$CV = \frac{\sqrt{\sum_{i=1}^{n}(x_i - \bar{x})^2}}{\bar{x}} \tag{4}$$

in which $x_i$ is the rainfall value in the center of the $i$th grid cell, in mm; $\bar{x}$ is the average rainfall of all grid cells over the observed scene, in mm; $n$ is the number of grid cells.

#### 3.2.2. Global Moran's I and Local Moran's I

Compared with CV, spatial autocorrelation indicators can identify the relationship of variables between different locations based on the regionalized variable. Global Moran's I [57] (Moran's I), Local Moran's I (Local Indicators of Spatial Association, LISA) [41] are both statistical methods for detecting the presence of spatial autocorrelation. Moran's I value only varies from −1 to 1. Moran's I > 0 indicates a positive spatial autocorrelation. A higher value indicates a higher autocorrelation. Moran's I < 0 indicates a negative spatial autocorrelation, and 0 represents randomness. A positive

spatial autocorrelation means that the adjacent observations in geography have similar characteristics, in other words, high/low rainfall values are collocated with similar high/low ones.

$$I = \frac{n}{\sum_{i=1}^{n}\sum_{j=1}^{n} w_{ij}} \frac{\sum_{i=1}^{n}\sum_{j=1}^{n} w_{ij} z_i z_j}{\sum_{i=1}^{n} z_i^2} = \frac{n}{\sum_{i=1}^{n}\sum_{j=1}^{n} w_{ij}} \frac{\sum_{i=1}^{n}\sum_{j=1}^{n} w_{ij}(x_i - \bar{x})(x_j - \bar{x})}{\sum_{i=1}^{n}(x_i - \bar{x})^2} \tag{5}$$

Considering grid cells as the spatial unit, and $x_i$, $x_j$ are the rainfall value at $i$th, $j$th grid cell, in mm; $n$ is the number of grid cells. $w_{ij}$ specified in Equation (5) is an element in a spatial weight matrix $W$:

$$W = \begin{bmatrix} w_{11} & w_{12} & \cdots & w_{1n} \\ w_{21} & w_{22} & \cdots & w_{2n} \\ \vdots & \vdots & \ddots & \vdots \\ w_{n1} & w_{n2} & \cdots & w_{nn} \end{bmatrix} \tag{6}$$

Since the regular distribution of the grid cells, we calculate the spatial weight matrix in the 0/1 weighting, where $w_{ij} = 1$, if $i$th and $j$th cells are contiguous, and $w_{ij} = 0$ otherwise. Note that grid cells in the diagonal position are not contiguous (Figure 2).

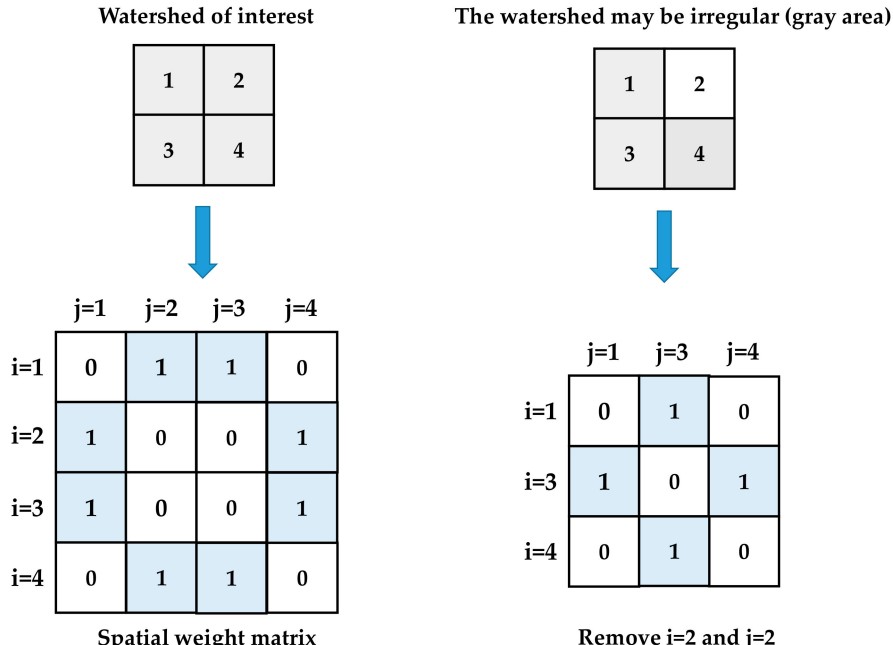

**Figure 2.** The calculation process of the spatial weight matrix.

While Moran's I [57] can only provide a single value to make a judgment about the spatial autocorrelation of the whole observed areas, Anselin [41] proposes a new technique in local, which describes the collocated location of the similar adjacent observations. The values of LISA are not limited to $[-1, 1]$. The relationship between Moran's I and LISA is shown below:

$$I = \frac{n^2}{\sum_{i=1}^{n}\sum_{j=1}^{n} w_{ij}} \frac{(x_i - \bar{x})\sum_{j=1}^{n} w_{ij}(x_j - \bar{x})}{\sum_{j=1}^{n}(x_j - \bar{x})^2} \tag{7}$$

$$\sum_{i=1}^{n} I_i = nI \tag{8}$$

### 3.2.3. Semi-Variance Analysis

To identify the spatial structure of rainfall variations and enrich the understanding of the range of rainfall clusters in different directions, the semi-variance analysis of geostatistics is taken as indicated here. The following parameters are used in it: nugget ($C_0$), partial sill ($C$), sill ($C_0 + C$), range ($\alpha$) (Figure 3). $C_0$ depicts the random spatial variance of the rainfall observations caused by uncertainties rather than rainfall data, while $C$ means structural spatial variance is caused by rainfall intrinsically. $C_0 + C$ is the total variation of the spatial heterogeneity. Based on them, the ratio of $C_0/C_0 + C$ (Nugget-Sill Ratio, NSR) [58] can also be used as the measure of spatial variance structures. NSR < 25% indicates a strong spatial autocorrelation of rainfall observations. 25% < NSR < 75% indicates a moderate spatial autocorrelation in space. NSR >75% indicates a weak spatial autocorrelation. $\alpha$ represents the maximum spatial distance of rainfall autocorrelation, that is to say, rainfall beyond this range is considered to have no spatial correlation [42].

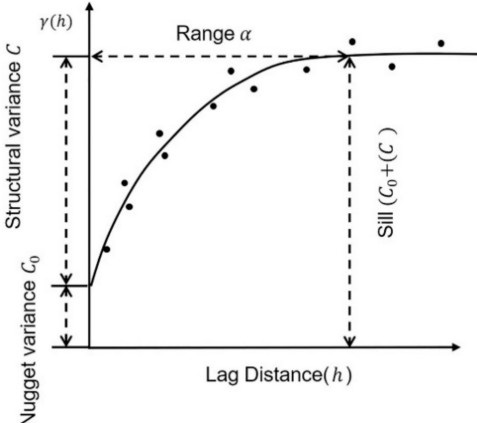

**Figure 3.** A schematic diagram of the semi-variance analysis with its key parameters [58].

Semi-variance is given by the following expression:

$$\gamma(h) = \frac{1}{2N(h)} \sum_{i=1}^{N(h)} [Z(y_i) - Z(y_{i+h})]^2 \tag{9}$$

in which, $Z(y_i)$, $Z(y_{i+h})$ is the rainfall values at the location of $y_i$ and $y_{i+h}$, respectively, in mm; $N(h)$ is the number of pair of points separated by a distance $h$ [59]; $h$ is the lag distance, which we define regular grid cells as the spatial unit, and one gridding length (10 km) considered as one $h$ in this paper. In general, owing to fewer grid pairs in large $h$, the $\gamma(h)$ is not statistically reliable if $h$ longer than 1/2 of the maximum distance in study areas [60].

The aspherical model is selected to fit the empirical semi-variogram, and the correlation coefficient ($R^2$) is calculated to assess the goodness of fit. The spherical is defined in Equation (10), and Equation (11) is the expression of the correlation coefficient.

$$\gamma(h) = \begin{cases} 0 & h = 0 \\ C_0 + C\left(\frac{3}{2}\frac{h}{\alpha} - \frac{1}{2}\frac{h^3}{\alpha^3}\right) & 0 < h < \alpha \\ C_0 + C & h > a \end{cases} \tag{10}$$

$$R^2 = 1 - \frac{\sum_{i=1}^{n}\left(\hat{y}^{(i)} - y^{(i)}\right)^2}{\sum_{i=1}^{n}\left(\overline{y} - y^{(i)}\right)^2} = 1 - \frac{MSE\left(\hat{y}^{(i)}, y\right)}{Var(y)} \tag{11}$$

where $\hat{y}^{(i)}$ are values in the fitting curves; $y^{(i)}$ are values of $\gamma(h)$ within different $h$, and $\overline{y}$ is mean of $y^{(i)}$.

## 4. Results and Discussion

### 4.1. Temporal Characteristics of Rainfall Series

The magnitude characteristics of the Rx3h are examined in Figure 4 and Table 2. The Rx3h ranges from 11 mm to 99 mm over the four cities. The minimum is observed in BJ, whereas the maximum in SZ. It shows that the magnitude of extreme rainfall increases from north China to south China, which is paired with the geographical locations of the four cities. The pattern of magnitude distribution can be illustrated by the length of whiskers between the median and the maximum/minimum line segment in the box plot (Figure 4). The rainfall in SZ, GZ, and SH is more concentrated on the larger side of the value (median to the maximum side), and the distribution shows the right skew. More heavy rainfall is prone to occur in a short duration in these cities. The distribution of Rx3h magnitude is relatively concentrated in BJ, around 18.9 mm. Through the length of the box or standard deviation in Table 2, The higher variation of Rx3h magnitude is found in SZ, followed by the GZ, SH, and BJ. Though the lowest variation of magnitude, a few outliers that occur in BJ reached 40 mm, and 42 mm which should also be given attention.

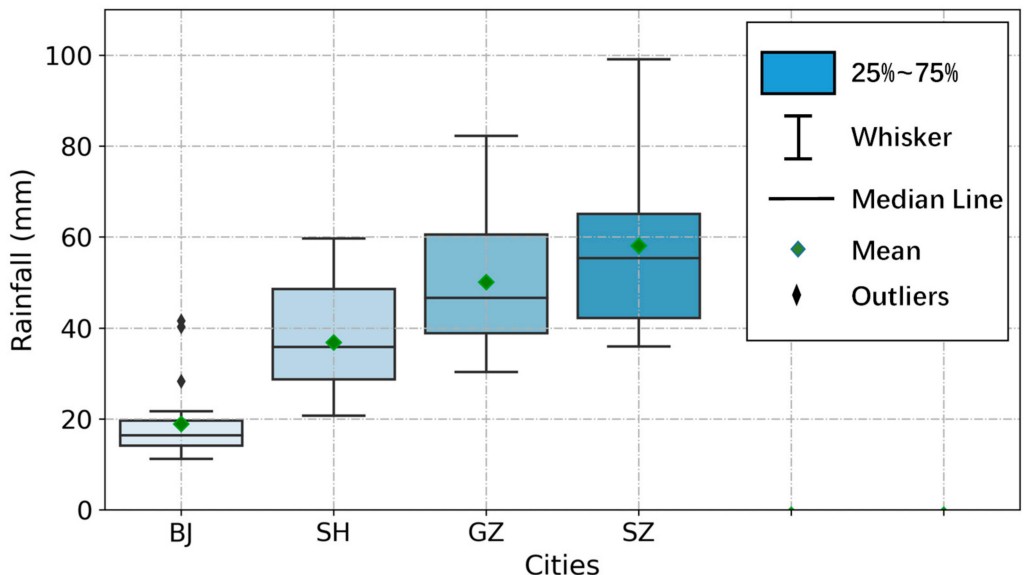

**Figure 4.** A box and whisker plot of Rx3h magnitude in study cities (The whisker range within 1.5IQR, where IQR = Q3 − Q1, Q1 means 25th percentile, Q2 means 50th percentile, Q3 means 75th percentile).

**Table 2.** Information of Rx3h series in study areas.

| Cities | Maximum (mm) | Mean (mm) | Minimum (mm) | Standard Deviation |
|--------|--------------|-----------|--------------|--------------------|
| SH | 59.7 | 36.8 | 20.8 | 11.2 |
| BJ | 41.6 | 18.9 | 11.2 | 8.4 |
| GZ | 82.3 | 50.1 | 30.3 | 14.1 |
| SZ | 99.1 | 58.1 | 35.9 | 18.8 |

The linear trends are analyzed by the M-K test and Sen's Slope method, as shown in Figure 5. The M-K analysis shows that the Rx3h series have no significant trend in four cities. By Sen's slope estimator, it shows that the tendency rates in four cities are smaller than 1 mm/a. It can be found that there is no pronounced change in the magnitude of short-duration extreme rainfall in these megacities during 2000–2020.

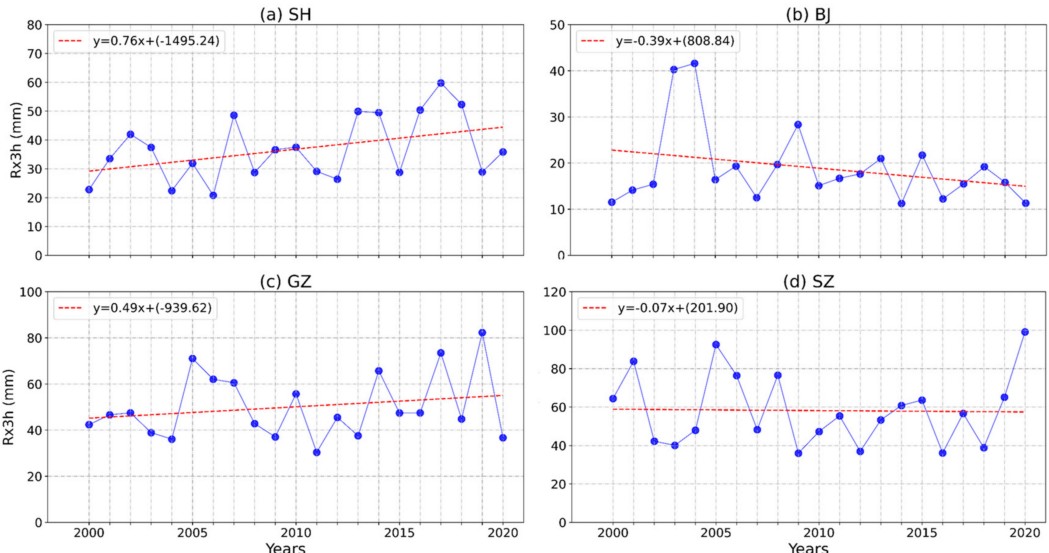

**Figure 5.** Trend analysis of Rx3h series in study cities (The blue dots are the values of Rx3h, and the red lines are the trend lines calculated by Sen's slope method).

The occurrences of the 3-h extreme rainfall in the four cities show significant seasonality (Figure 6). The probability density function of storm occurrence is computed using the 2-D Gaussian kernel based on the date and time when Rx3h occurred. In SH, Rx3h events concentrate around 10:00–16:00 of July and August. The maximum probability of Rx3h occurrence is in August with the most occurrence number and the most rainfall. The accumulations of rainfall in August reached 291.6 mm accounts for 39.24% of the total rainfall of the series. In BJ, Rx3h events are concentrated in the morning (around 8:00–12:00) of July and August. 57% of the Rx3h events in BJ occur in July, contributing to 66.3% of the total rainfall. The Rx3h in GZ and SZ concentrate in April–June because the rainy season in China starts earlier in the south than in the northern areas. Rx3h most often occurs in the morning in GZ while in the early nighttime and morning in SZ. It has the highest probability of occurrence in May both in GZ and SZ, and the accumulation of Rx3h in SZ is larger, reaching 499 mm. Compared with SZ, Rx3h in BJ and GZ concentrate more over the hourly timeframe. Considering the greater rainfall magnitudes in GZ, it is of greater concern.

*4.2. Spatial Heterogeneity of Extreme Rainfall*

The spatial distribution of extreme storms varies in the four cities. The map of mean Rx3h during the 2000–2020 period in study areas is shown in Figure 7. In SH and GZ, the largest rainfall is located around the city center and decreases toward the watershed boundaries. In BJ, the rainfall increases from the west to the east and clusters in the northeast corner which is a flat plain. In SZ, the mean of Rx3h clusters in the north-central and decreases toward the south. The results show that the spatial distribution of extreme rainfall is non-uniform in the four cities. In this section, hence, the rainfall spatial heterogeneity will be quantitatively identified and assessed.

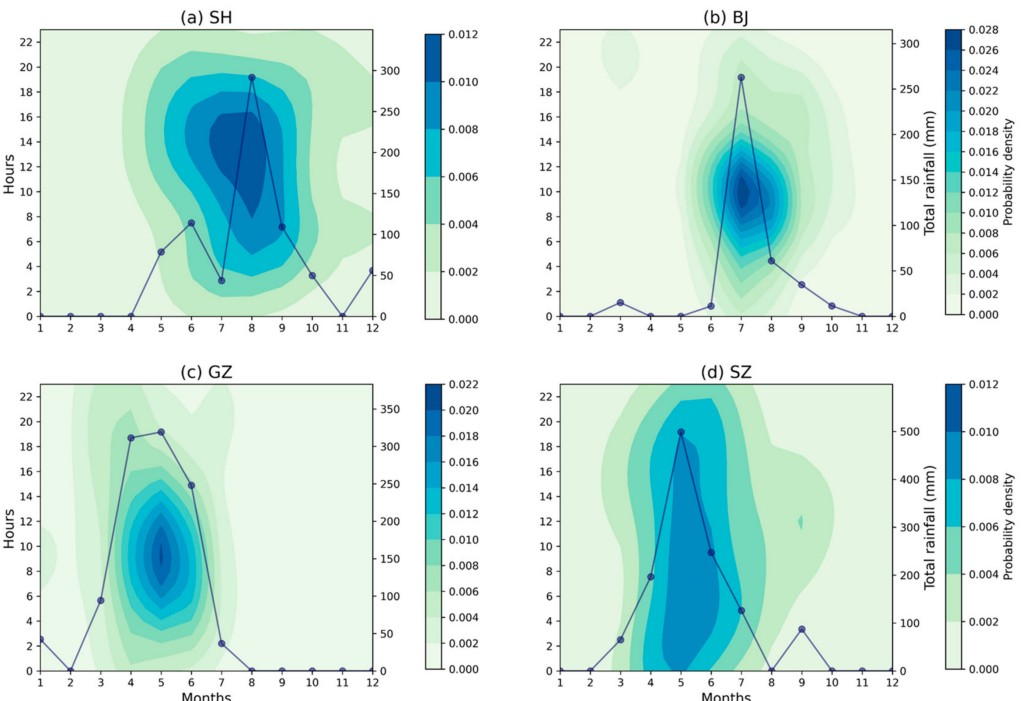

**Figure 6.** Diagram of probability density distribution to analyze the Rx3h occurrence (Point plots in figure means the total rainfall in months).

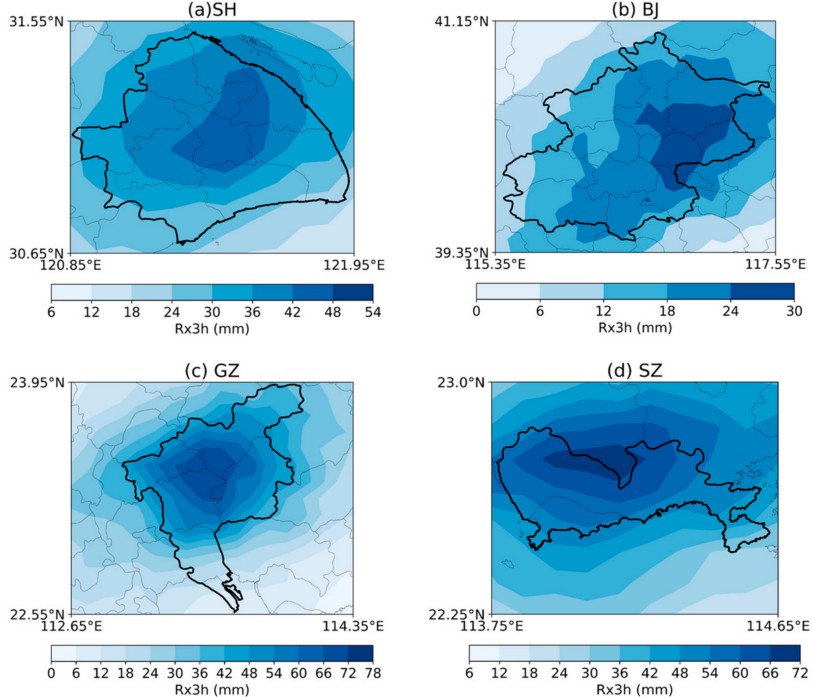

**Figure 7.** Maps of mean Rx3h in SH, BJ, GZ, and SZ (2000–2020).

### 4.2.1. Rainfall Clustering Analysis

The spatial variability of Rx3h is examined through the analysis of CV and Moran's I (Table 3). The Rx3h in the four cities show four different patterns of spatial clustering. The scatter diagrams of Moran's I (Figure 8) demonstrate the concrete features of clustering. The scatter falls in the first, second, third, and fourth quadrants, indicating high values aggregation, the low value itself surrounded by

high values, the high value itself surrounded by low values, and low values aggregation, respectively. Z-score, which is the result of hypothesis testing, with values greater than 1.96, indicates the spatially clustered of rainfall is reliable statistically [40].

**Table 3.** The results of Moran's I and CV in four study cities.

| Cities | Moran's I | CV | Z-Score |
|---|---|---|---|
| Shanghai | 0.472 | 0.473 | 5.16 |
| Beijing | 0.621 | 0.487 | 12.109 |
| Guangzhou | 0.628 | 0.635 | 8.00 |
| Shenzhen | 0.215 | 0.631 | 1.771 |

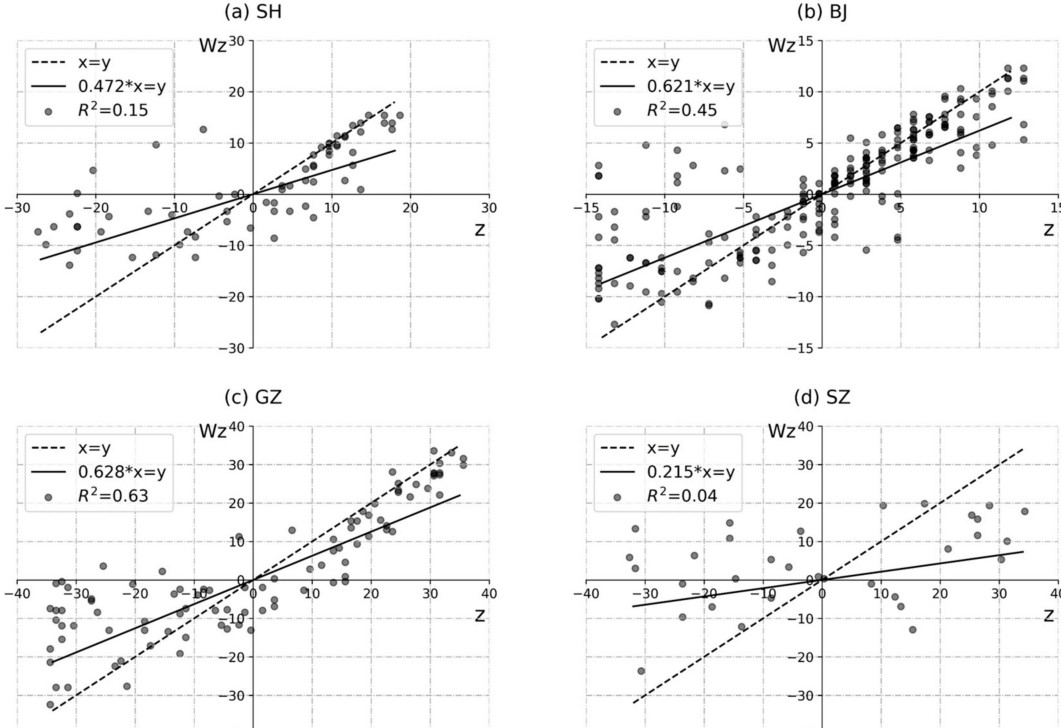

**Figure 8.** Scatter diagrams of Moran's I in four study cities.

In BJ, the high Moran's I and low CV indicate that Rx3h is highly clustered in space. However, there is relatively little difference in rainfall magnitude between the clustered area and the rest part. Contrary to BJ, SZ has low Moran's I (positive) and high CV, indicating that the clustering feature of Rx3h here is relatively insignificant, but the variability of rainfall magnitude is remarkable. In other words, the variability in space is more complex in SZ, which is also illustrated in the complex scatter distribution in Figure 8d. In GZ, Moran's I and CV are both high, showing both the noticeable clustering of spatial distribution and variability of rainfall magnitude. The scatters concentrated on the line "x = y" in Figure 8c address the highly clustered pattern. Extreme storms concentrate in local areas in GZ with large rainfall magnitude, where likely has a higher risk of rainfall-induced flooding. Compared with GZ, Moran's I and CV were both low (positive) in SH, showing both the moderate clustering of rainfall distribution and variability of rainfall magnitudes. The above results highlight that GZ has the largest rainfall spatial heterogeneity, while SZ has the smallest one.

The locations of the local spatial clusters of study cities are identified by LISA analysis [41]. All the clusters show a monocentric pattern in study areas (Figure 9). The comparison to the results in Figure 7 proves the reliability of the results in LISA analysis. In SH, the Rx3h clusters in the central city area and southern suburbs with a trend of decreasing toward the watershed boundaries. It may be associated

with the urban heat island effect and sea-land breeze circulation. In BJ, Rx3h clusters locate in the east of BJ and show a north-to-south spatial gradient across the plain. With the mountainous terrain to the west and the urban growth from the southeast, extreme rainfall appears to be positively influenced by mountain-plain circulations. In GZ, the Rx3h events cluster in the central area of the city with a decreasing spatial trend to other portions, which is closely linked to the topography of high in the north, low in the south, and tropical weather system. In SZ, the spatial clusters are located on the north side, especially in the northwest. Affected by both its elongated shape and significant southwest monsoon, the rain belt is generally across the whole city. The results imply that local spatial clusters can be influenced by climate, topography, and the pattern of urban growth.

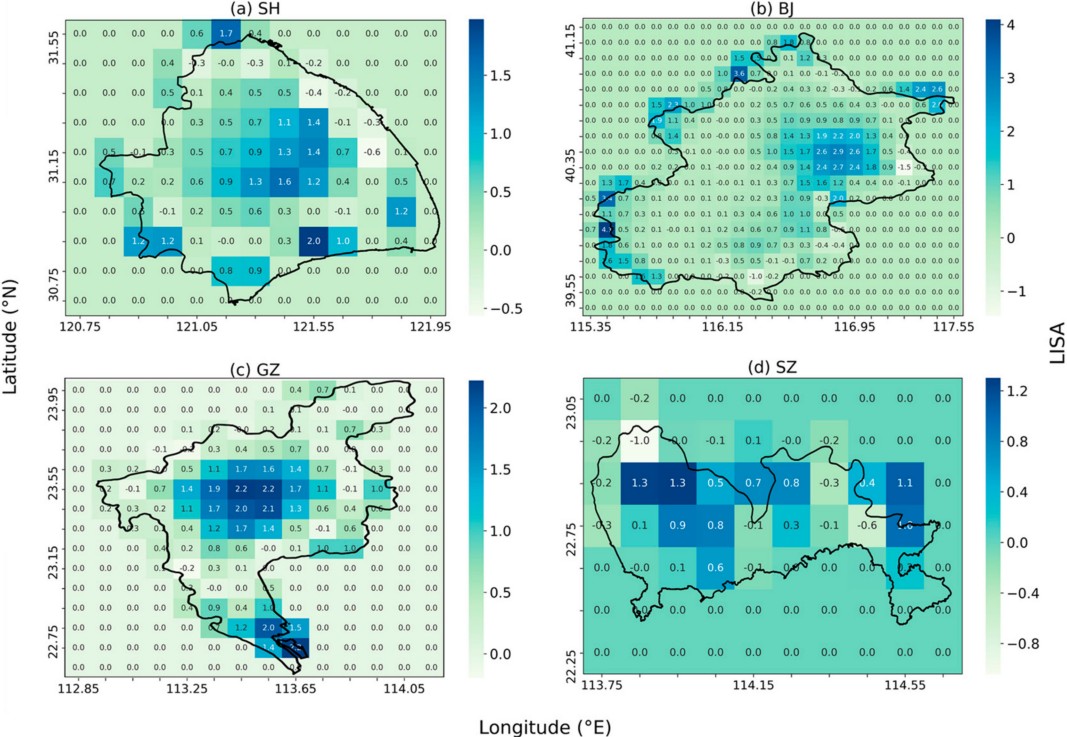

**Figure 9.** Analysis of locations of clusters (results in LISA) for four study cities. (Gridded cells which outside the scope of study areas are covered by 0 values, zero represents randomness in spatial autocorrelation analysis).

The range of rainfall clusters is examined by semi-variance analysis. Generally, the semi-variogram is an increasing curve of the lag distance ($h$), and it may reach a sill or increase indefinitely as $h$ increases. We use regular grids as the analysis unit in this study. For example, if SH includes 12 grid cells in longitude and 6 grid cells in latitude, a statistically reliable semi-variance analysis can be carried out within $1 \leq h \leq \frac{1}{2} \times 12$.

The semi-variance is first computed without considering the change of rainfall in different directions (Figure 10). The spatial distribution of short-duration extreme rainfall in urban regions shows pronounced spatial aggregation. Variograms of the four cities can reach a stable value (sill) within the range of lag distance ($\alpha$), which demonstrates that each range of the Rx3h clusters is smaller than the city scale. In SH, for instance, the distance where the variogram reaches the sill is 4.84 ($\alpha = 4.84$), indicating the Rx3h extends to a scale of 48.4 km × 48.4 km. The results of other cities are presented in Table 4.

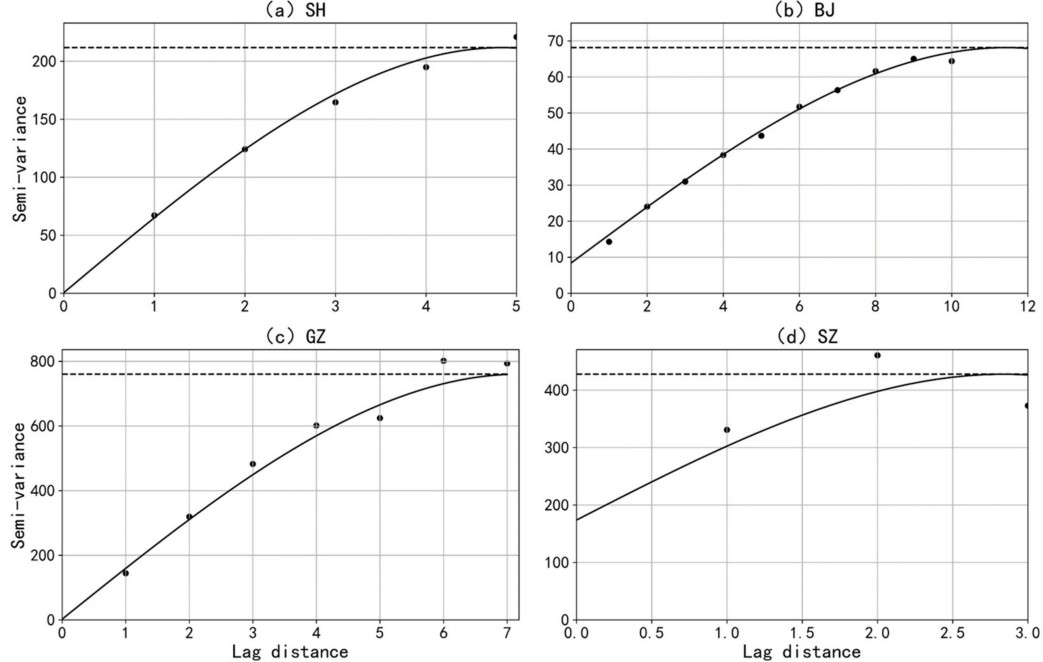

**Figure 10.** Semi-variance analysis for four cities without considering the change of rainfall in different directions.

**Table 4.** Key parameters in semi-variance analysis for four cities.

| Cities | Orientation | $C_0$ | $C_0 + C$ | $\alpha$ | $\frac{C_0}{C_0 + C}$ | $R^2$ |
|---|---|---|---|---|---|---|
| Shanghai | - | 0.487 | 211.753 | 4.840 | 0.2% | 0.893 |
| | E-W | 3.185 | 164.353 | 5.226 | 1.9% | 0.993 |
| | N-S | 20.677 | 290.818 | 5.872 | 7.1% | 0.994 |
| Beijing | - | 8.369 | 68.141 | 11.43 | 12.3% | 0.997 |
| | E-W | 2.554 | 94.107 | 15.532 | 2.7% | 0.999 |
| | N-S | 10.574 | 53.445 | 7.991 | 19.8% | 0.996 |
| Guangzhou | - | 9.469 | 792.589 | 7.669 | 1.2% | 0.976 |
| | E-W | 0.679 | 747.718 | 5.798 | 0.09% | 0.974 |
| | N-S | 15.205 | 1191.336 | 11.918 | 1.3% | 0.993 |
| Shenzhen | - | 173.800 | 427.700 | 2.840 | 26.6% | 0.248 |
| | E-W | 67.589 | 414.419 | 4.098 | 16.3% | 0.987 |
| | N-S | 69.477 | 679.700 | 1.980 | 9.9% | 0.999 |

The other key parameters in Table 4 also reflect different characteristics of spatial data variance. The small nugget variance ($C_0$) in SH and GZ indicates that it is less affected by other random factors (such as human factors, measurement errors, or others) [42,58,61]. On the contrary, $C_0$ values are larger in BJ and SZ, with the nugget-sill ratio ($C_0/C_0 + C$, NSR) even reaching 12.3% and 26.6%, respectively. It implies that the uncertainty is significant and complex for the spatial variability in BJ and SZ. The results of $R^2$ show the goodness of fit for the spherical model performed well in SH, BJ, and GZ. The low $R^2$ in SZ is likely associated with fewer fit points and no-significant spatial autocorrelation.

To further examine the change of rainfall variability in different directions, the north–south and east–west directions are selected to calculate the semi-variance (Figure 11). The spatial distributions of annual max 3-h rainfall in the four cities are all anisotropic. In SH, the spatial variability of Rx3h in the north-south direction of the city is greater than that in the east-west direction, and the range of the rainfall clusters shows a balanced pattern between transverse and longitudinal directions (52 km × 58 km). Within 50 km, the spatial variability in GZ was more influenced by rainfall in the east–west than north–south. The semi-variance in the east-west direction reaches the sill at about

58 km ($\alpha = 5.798$), while there is still a growing trend in north-south. It indicates that the observations still have spatial autocorrelation within the predetermined distance. The boundary of the rainfall clusters may go beyond the predefined spatial extent. Similarly, it can be proved that the Rx3h in GZ is more manifested as a longitudinal (north–south) clustering pattern. Rx3h clusters in BJ and SZ both show a transverse (east–west) clustering pattern.

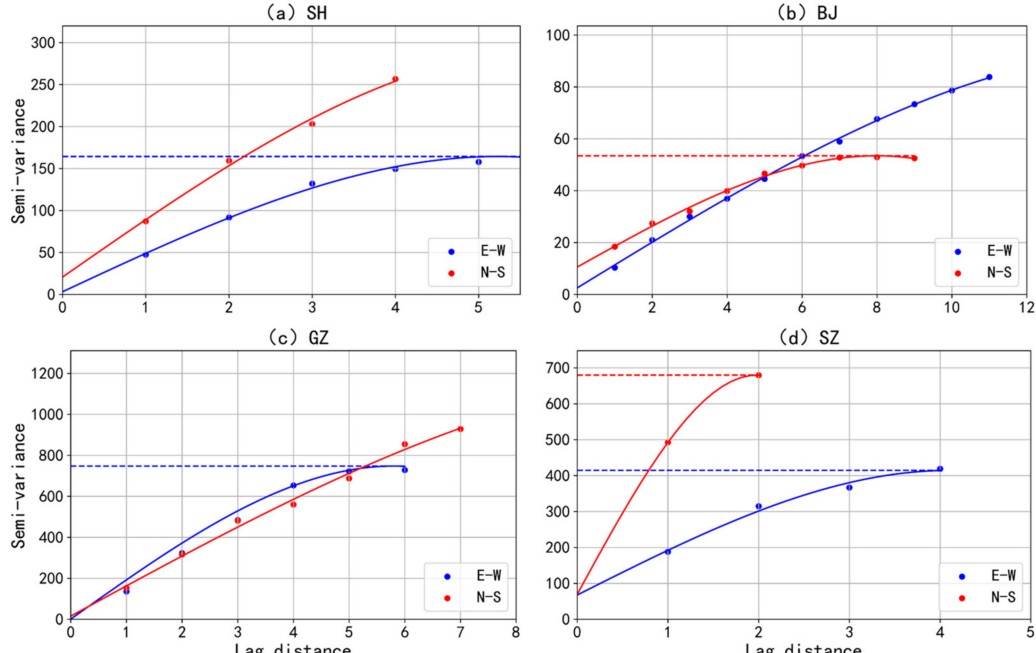

**Figure 11.** Semi-variance analysis for two directions: east–west and north–south.

A radio is used to assess the size of the aggregation area and it is denoted by the following: $R_\alpha = \frac{\alpha_{N-S}\alpha_{E-W}}{N_{lat}N_{lon}}$. In which $\alpha_{N-S}$, $\alpha_{E-W}$ are the range ($\alpha$) obtained from the two directions; $N_{lat}$, $N_{lon}$ are the number of the grid cells in latitude and longitude. It can be found that the area of rainfall clusters in GZ is the largest with a ratio of 37.97%. The area of rainfall clusters in SZ is the smallest with a ratio of 12.88%. SZ and BJ have moderate ratios of 28.41%, 28.40%.

### 4.2.2. Distribution of Rainfall Centroids

Rainfall-movement strongly impacts surface flow and peak discharges [62,63], making it an important issue in the extreme rainfall study. Spatial heterogeneity is manifested not only in the magnitudes and distribution of areal mean rainfall but also in the moving of the rainfall centroids [23]. Rainfall centroids indicate the locations of the maximum rainfall in a certain duration, which should be alarmed if there is excessive or prolonged rainfall at the centers.

The location of Rx3h centroids is demonstrated in Figure 12. The dot colors indicate the magnitude of rainfall. If the rainfall centroids occur at the same location more than once, the rainfall will be accumulated and shown in Figure 12. In SH, the rainfall centroids cluster in the central portions and tend to distribute along the Huangpu River. The rainfall centroids in BJ distribute in the northeast plain and central urban area (southeast area). It spreads out to the east, with only two rainfall events occurring in the same location. In GZ, most of the rainfall centroids cluster in the center of the city. In SZ, centroids distribute on the north side with repetitions in the same location. It should be noted that there is a large rainfall event of about 202 mm occurs in the northeast, even more than the accumulation of four rainfall events at the center.

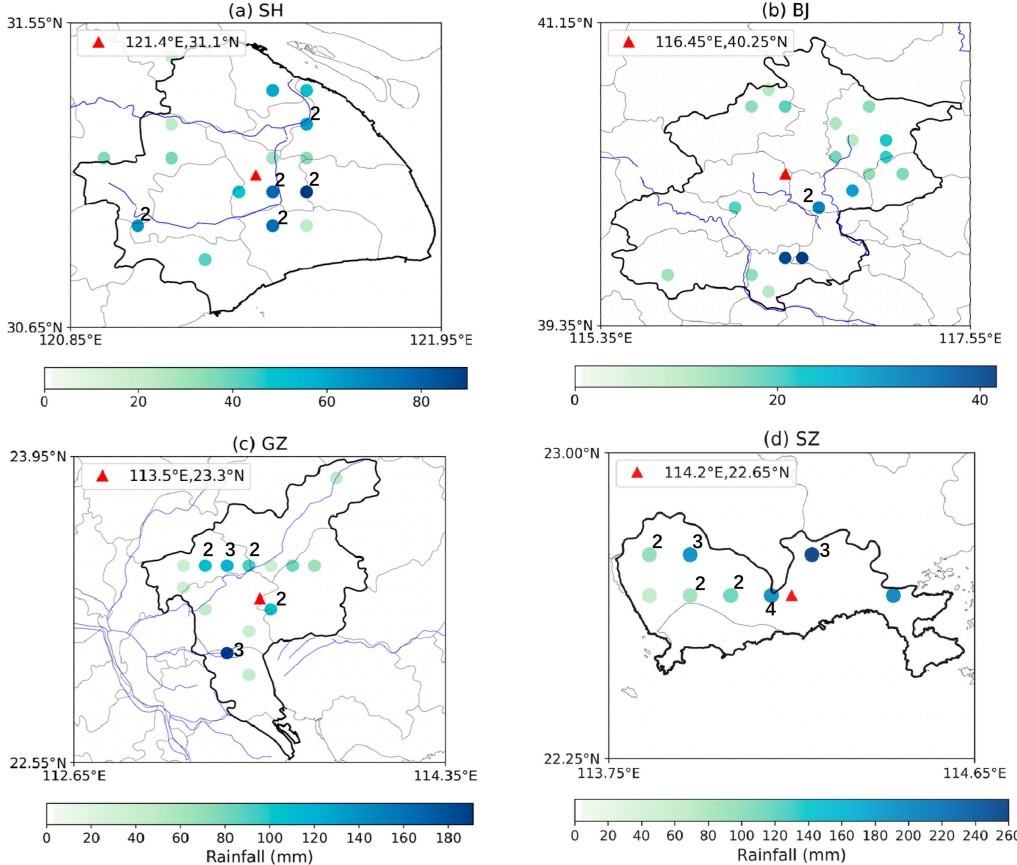

**Figure 12.** Locations of Rx3h centroids during 2000–2020 (The dot locations are where the Rx3h centroids occur in the cities, and the dot colors indicate the magnitudes of rainfall. The different numbers reveal the occurrence number of the centroids at the marked location, and the number not marked is one time. Geographic location centers in four cities are represented by a red dot. The blue lines for the nearby river).

The wind rose is used to illustrate the relative position of rainfall centroids and geographic location centers (Figure 13). In SH, compared to the geographic location centers, the occurrence probability and magnitudes of rainfall on the east side were both larger than that on the west side. In BJ, 80% of rainfall centroids with a magnitude ranging from 20 to 52 mm show a pattern of clustering in the east. As mentioned above, it is likely linked to the mountain-plain topography. In GZ, the maximum magnitude of the rainfall centroids reaches 68–84 mm, which is distributed in the southwest, northwest, and northeast of the geographical center. In SZ, centroids with the highest probability of occurrence are located in the north while the maximum magnitudes of rainfall occurred close to the northwest. Generally, the distribution of rainfall centroids and rainfall clusters is consistent in space.

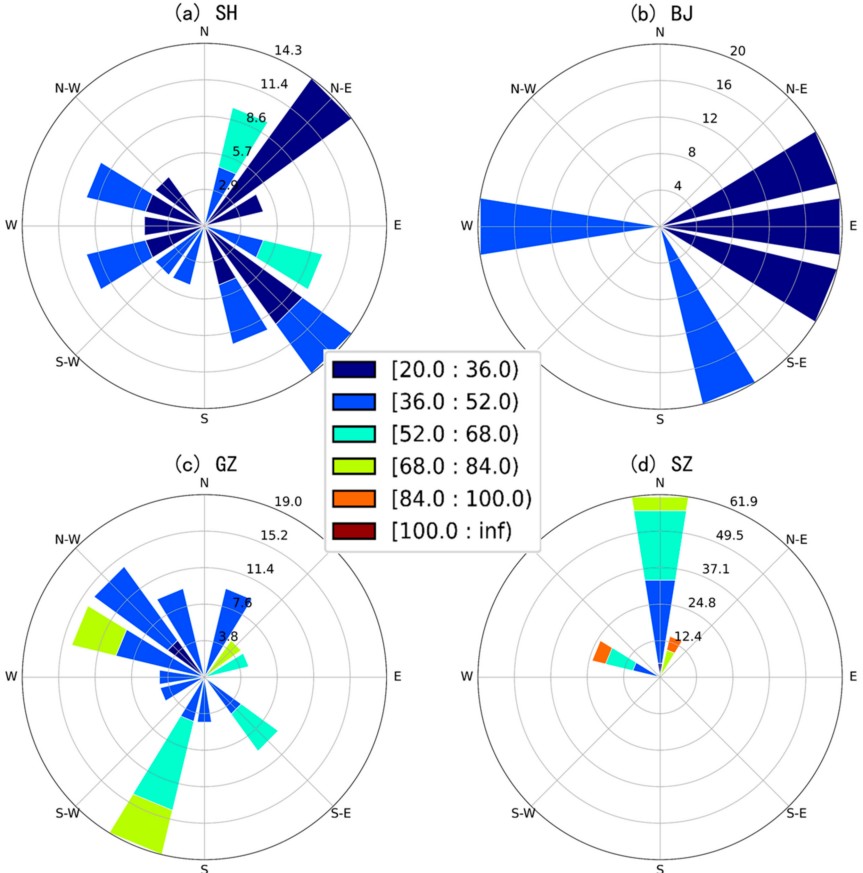

**Figure 13.** The wind rose to analyze the relative positions between rainfall centroids and geographic location centers (The relative positions between rainfall centroids and geographic location centers are divided into eight directions. The colors of the ribbons indicate the rainfall magnitudes of rainfall centroids, and the length of it indicates the occurrence probability of these magnitudes).

The spatio-temporal information of rainfall centroids is extracted per time step to examine the path of rainfall events. For a 3-h time scale, 6 centroids can be obtained from the 3-h rainfall process (temporal resolution = 30 min). The movement of centroids can be simply described by comparing the locations of the first and last centroid. CV is computed by the rainfall magnitude at 6 centroids (Table 5).

**Table 5.** A dynamic analysis of the Rx3h centroids at a 3-h time scale (temporal resolution = 30 min). "Frequency" represents the percentage of the events during 21-years.

| Cities | Mean of Moving Speed (km/h) | Most Frequently Moving Angles (Frequency) | CV |
|--------|------------------------------|--------------------------------------------|-----|
| SH | 12.3 | 270–315 (28.6%) | 0.27 |
| BJ | 24.7 | 225–270 (42.9%) | 0.22 |
| GZ | 13.7 | 270–315 (28.6%) | 0.24 |
| SZ | 11.7 | 0–45 (23.8%), 90–135 (23.8%) | 0.28 |

The results of centroid movements are shown in Table 4. During 2000–2020, 42.9% of rainfall centroids in BJ move toward the southwest of the city, which are paired with the growing trend of urbanization reported by Wang [11]. Greater urban flood risk may be increased under the similar direction of urban expansion to the location of clustered rainfall. In SH, 28% of rainfall centroids move toward the southeast coast. Centroids in GZ clusters in the central portion of the city and move toward the southeast in a relatively small range. The rainfall centroids in SZ mostly cluster in the north

portion. The two most common directions of movement in SZ are along the northeast–southwest and northwest–southeast which is closely related to the South China Sea monsoon.

*4.3. Discussion*

4.3.1. Influence of Urban Growth on Spatial Heterogeneity of Rainfall

We compare the change of rainfall spatial heterogeneity with that of urbanization to examine the impact of urbanization on extreme rainfall distribution. One of the most important features of urbanization is the change of land surface from a permeable area to an impervious one (roads, plazas, airports, and parking lots). Based on maps of land use (Figure 1), the impervious surface coverage is calculated as the impervious surface area/total area. Generally, the impervious surface area of the four cities shows an increasing trend during the last 20 years (Table 6). In SH, the impervious surface coverage increases from 15.21% to 55.66% with an average rate of 2.53% per year. The rate of increase from 2015 to 2017 highly reaches 9.75% per year, implying fast urbanization during the two years. The increasing rate of GZ and SZ is 0.69% and 0.52% per year from 2001 to 2017, respectively. Both experience more rapid urbanization in 2015–2017 with an increasing trend of 4.98% and 1.120% per year, respectively. In contrast, the change of impervious surface coverage in BJ is the slowest, with 0.20% per year during 2001–2017. SH shows the largest increase in urbanization, followed by GZ, SZ, and BJ.

**Table 6.** Changes of impervious surface area in the study areas.

| Cities | 2001 (%) | 2015 (%) | 2017 (%) | The Annual Rate of Increase (2001–2015) (% Per Year) | The Annual Rate of Increase (2015–2017) (% Per Year) | The Annual Rate of Increase (2001–2017) (% Per Year) |
|---|---|---|---|---|---|---|
| SH | 15.22 | 36.16 | 55.66 | 1.50 | 9.75 | 2.53 |
| BJ | 12.34 | 13.69 | 15.46 | 0.13 | 0.89 | 0.20 |
| GZ | 13.19 | 14.48 | 24.44 | 0.09 | 4.98 | 0.69 |
| SZ | 29.21 | 35.14 | 37.54 | 0.42 | 1.12 | 0.52 |

The annual change of the Moran's I in four cities is demonstrated in Figure 14. Through the M-K test, Moran's I in SH shows a significant increasing trend. BJ and GZ both show no obvious trend while SZ even shows a decreasing trend. In 2001, 2015 and 2017, the Moran's I in SH are 0.518, 0.659, and 0.694, respectively. The most rapid urbanization and the most significant increase in Moran's I in SH imply that urban growth impacts the distribution of regional extreme storms.

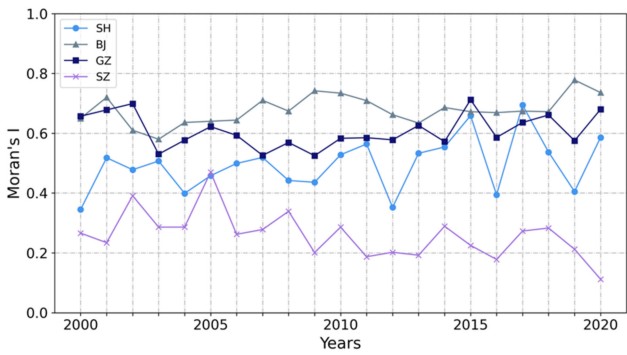

**Figure 14.** Line charts of annual change in Moran's I based on 3-h extreme rainfall events in study areas.

Results of land use maps, LISA maps, and semi-variance analysis in 2001, 2015, and 2017 in SH are shown in Figure 15. Urban growth in SH starts from the northeast corner and gradually extends to the city boundary (Figure 15a). LISA maps (Figure 15b) show that the rainfall clusters move from the central areas of the city to the northeast. Correspondingly, the impervious area also increases from the city center to the northeast. This result implies that the location of 3-h extreme rainfall in

SH is impacted by urbanization, tending to the most urbanized area. There is no distinct change in the scale of rainfall clusters during the 2001–2017 period (Figure 15c) with the range remaining about 4–5 km. These results capture the changes in impervious surface area and rainfall clustering in SH, highlighting the impact of urban growth on spatial heterogeneity of rainfall.

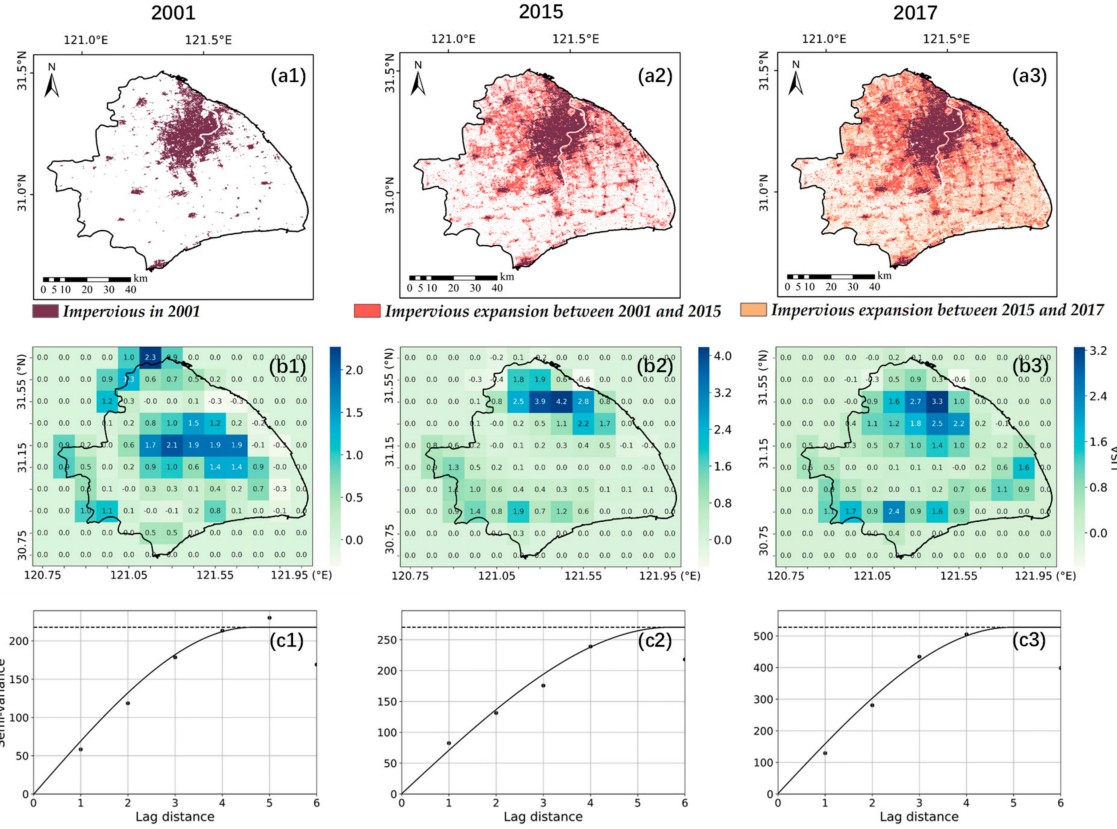

**Figure 15.** Influence of urban growth on spatial heterogeneity of 3-h extreme rainfall in SH: (**a**) Land use maps (only focus on impervious) in 2001, 2015, 2017, (**b**) LISA maps in 2001, 2015, 2017, (**c**) Semi-variance analysis in 2001, 2015, 2017.

### 4.3.2. Influence of Rainfall Spatial Heterogeneity on Design Storm

The impact of extreme rainfall spatial heterogeneity on rainfall frequency analysis is examined through the stochastic storm transposition method (SST). SST is an alternative approach for assessing the frequency of extreme rainfall (so-called "design-storm") and flooding, which has been applied in the United States [2,64] and China [26]. The reader is directed to Zhou et al. [33] and references therein for further details. The main idea of SST is to use regional probabilistic resampling combined with storm geospatial translation (transposition) to estimate local-scale extreme event frequency. The geospatial transposition can be set as uniform or non-uniform based on the spatial properties of the transposition area, providing a viable approach to assess the spatial properties of storms. In this study, the Intensity-Duration-Frequency (IDF) curves in SH are estimated with rainfall spatial heterogeneity and without rainfall spatial heterogeneity. The comparison (Figure 16) shows that the impact on the frequency of extreme rainfall is different. The median IDF using non-uniform transposition results in higher estimates than that using uniform transposition for all the return periods. The difference is a little more significant in larger return periods. It demonstrates that regional spatial heterogeneity impacts short-duration rainfall estimation, especially for large return periods. Without considering the rainfall spatial heterogeneity will lead to underestimation of design storm. Spatial variability of rainfall is suggested to be taken into account in rainfall/flood calculations.

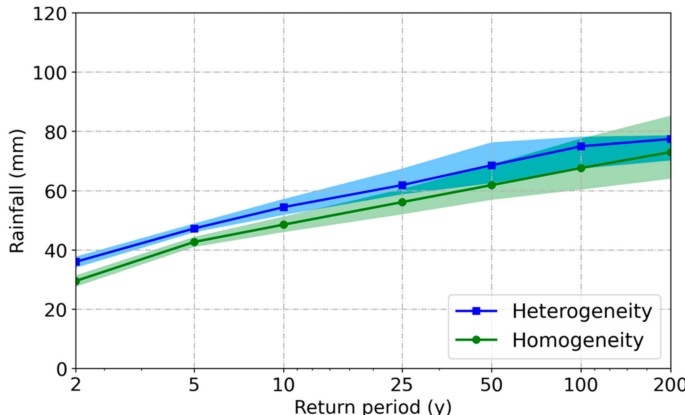

**Figure 16.** Comparison of Intensity-Duration-Frequency (IDF) curves uses non-uniform spatial transposition (heterogeneity) and uniform transposition (homogeneity) scenarios, which are calculated based on the SST method at a 3-h time scale. The blue/green areas indicate the spread of SST estimates with 90th and 10th quantiles as the solid lines.

## 5. Summary and Conclusions

In this study, we present a comprehensive framework to analyze the spatial heterogeneity of rainfall in four megacities in China. This framework focuses on spatial cluster characteristics of short-duration extreme rainfall. The locations of rainfall centroids and their variation under heterogeneous are also analyzed. Dynamic analysis based on land use maps tries to explore the relationship between spatial heterogeneity and urban growth in megacities. The study is based on 21 years of extreme 3-h rainfall events extracted from the high-resolution gridded rainfall dataset (resolution of 30 min in time, $0.1° \times 0.1°$ in space). A few interesting conclusions achieved as follows:

(1) Though there is no significant change in the magnitude of short-duration extreme rainfall, pronounced rainfall spatial heterogeneity can be found in the four megacities in China. SZ has the largest magnitude of rainfall, while GZ has the largest spatial heterogeneity. It was no significant positive correlation between magnitude and spatial heterogeneity of rainfall. The spatio-temporal analysis of extreme rainfall should attach importance to not only rainfall magnitude but also to the spatial heterogeneity.

(2) The short-duration extreme rainfall exhibit four different patterns of spatial variability in SH, BJ, GZ, SZ. The extreme storm in SH clusters in the central portion, keeping a balance between transverse and longitudinal directions. In BJ, extreme storm clusters in the northeast plain with relatively small rainfall magnitude and a smooth process. The extreme storm in GZ shows a highly clustered pattern in the central areas accompanied by large rainfall magnitudes. Rainfall in the east-west attributed more than north-south to spatial clustering here. The clustering feature in SZ is not as significant as other cities, while the variability of rainfall magnitude is remarkable. It shows a transverse clustering pattern on the north side of the city. Due to different climate, topography, and land use, spatial heterogeneity of rainfall shows different characteristics in these four cities.

(3) Urban growth plays a role in the change of rainfall spatial heterogeneity. SH is a city that shows the largest increase in urbanization and rainfall spatial heterogeneity. During its rapid urbanization in the last 20 years, there is an increasing trend in Moran's I in SH. The rainfall clusters in SH also tend to cluster in the most urbanized areas. Rainfall spatial heterogeneity cannot be neglected in design storms and thus in the design of urban flood mitigation measurements.

Through the quantitative evaluation of rainfall spatial heterogeneity in the four most important cities in China, these results provide fundamental support for urban rainfall design and the rational decision-making process of rainfall-induced flooding in urban areas.

**Author Contributions:** Q.Z.: conceptualization, methodology, formal analysis, visualization, writing—original draft preparation, S.L.: conceptualization, writing—review and editing, supervision. Z.Z.: conceptualization, methodology, writing—review and editing, supervision. All authors have read and agreed to the published version of the manuscript.

**Funding:** This research was funded by National Natural Science Foundation of China (51909191 and 51961145106).

**Acknowledgments:** The authors are grateful for the important support given by National Natural Science Foundation of China (51909191 and 51961145106). The authors would also like to acknowledge the helpful comments of all reviewers without which the quality of the paper could not be improved.

**Conflicts of Interest:** The authors declare no conflict of interest. The funders had no role in the design of the study nor in the interpretation of data or the writing of the manuscript.

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
