# Peer review of "Spatial Heterogeneity Analysis of Short-Duration Extreme Rainfall Events in Megacities in China"

_water, doi:10.3390/w12123364_

Round 1

Reviewer 1 Report

The problem considered in the paper relates to spatial variability of the short-duration extreme rainfall  on the area of four major cities in China. The Authors present detailed geostatistical analyses including the rainfall trends, spatial heterogeneity of extreme rainfalls on data with clustering patterns, distribution of movement of the rainfall centroids in the period of 21 years. Moreover, the Authors prove that the urban growth of megacities affect the spatial heterogeneity of extreme rainfall.  An interesting conclusion is the authors' recommendation for calculating design rainfall, taking into account their spatial heterogeneity, without which the values of rainfall may be underestimated. The result section is well written and it is  convincing. The results of the article can be used in spatial planning of cities and rational  decision-making process of local flood protection level.

The research paper is interesting and sounds scientifically it also matches aims and scope of Water journal.

I recommend to correct a few errors or ambiguities:

Fig. 1 suggestion to  remove (a) from the top of the drawing and change (b1) to SH, (b2) to BJ, etc. on the maps.

Line 146 to 159 – apply a uniform convention for writing the coefficients in the formulas and the content of the article wij or wi,j  

Equ. (6) arrange the position of wij inside the matrix W

Line 154 – unclear sentence, I suggest replacing it with wi,j = 0 for i=j

Equ. (7) I suggest that you do not use the multiplication sign x

Fig. 2 – C1 replace with C

Equ. (9) a replace with α

Fig. 3. I think the "Minimum value" line is too high and there is no lower whisker

Line 319 – remove the sign * from the equation

Line 385 replace Figure 6 with Figure 1

Reviewer 2 Report

This article provides an interesting framework for analyzing the spatial heterogeneity of short-duration extreme rainfall events in four megacities of China. The paper is interesting and well written in general. However, I would request the authors to respond to the following comments before it can be accepted for publication.

  • The background of the study needs further strengthening. The following statement includes the main argument of this paper: “The main challenge faced by many researchers is how to represent and quantify the degree of spatial heterogeneity”. However, this is not enough. Questions still exist why researchers face challenges in quantifying spatial heterogeneity of rainfall. The following relevant studies on China are available in the existing literature. The authors need to justify the novelty of this work over these similar works:
  • Spatiotemporal variations of precipitation regimes across Yangtze River Basin, China (DOI 10.1007/s00704-013-0916-y)
  • Multi-decadal spatial and temporal changes of extreme precipitation patterns in northern China (Jing-Jin-Ji district, 1960–2013) (https://doi.org/10.1016/j.quaint.2018.03.008)

Also, the authors could link the extreme rainfall events to pluvial flooding. The following study could be helpful:

https://www.sciencedirect.com/science/article/pii/S0048969720322646

  • The English language needs attention. For instance, Line 10-11: “there is a growing need to analyze the spatial heterogeneity accurately”. I believe the authors intended to say “spatial heterogeneity” of rainfall. Please clarify in the sentence. This comment is applicable for line 11-12 also.
  • Line 17: “pronounced variability of spatial heterogeneity”. In this sentence, “variability” and “heterogeneity” are used redundantly. Please correct the sentence.
  • Figure 7: Please show R2 values in the diagrams.
  • The conclusion section should include the policy implications of this work.
